# ARDEP, a Rapid Degenerate Primer Design Pipeline Based on *k*-mers for Amplicon Microbiome Studies

**DOI:** 10.3390/ijerph17165958

**Published:** 2020-08-17

**Authors:** Yueni Wu, Kai Feng, Ziyan Wei, Zhujun Wang, Ye Deng

**Affiliations:** 1Key Laboratory for Environmental Biotechnology, Research Center for Eco-Environmental Sciences, Chinese Academy of Sciences, Beijing 100085, China; ynwu_st@rcees.ac.cn (Y.W.); kaifeng_st@rcees.ac.cn (K.F.); zjwang_st@rcees.ac.cn (Z.W.); 2College of Resources and Environment, University of Chinese Academy of Sciences, Beijing 100049, China; 3Institute of Marine Science and Technology, Shandong University, Qingdao 266237, China; w.ziyan@163.com

**Keywords:** bioinformatic program, primer design, *k*-mer, analysis platform, primer assessment

## Abstract

The survey of microbial diversity in various environments has relied upon the widespread use of well-evaluated amplification primers for taxonomic marker genes (e.g., prokaryotic 16S and fungal ITS). However, it is urgent to develop a fast and accurate bioinformatic program to design primers for microbial functional genes to explore more mechanisms in the microbial community. Here, we provide a rapid degenerate primer design pipeline (ARDEP) based on the *k*-mer algorithm, which can bypass the time-consuming step of sequence alignment to greatly reduce run times while ensuring accuracy. In addition, we developed an open-access platform for the implementation of primer design projects that could also calculate the amplification product length, GC content, Annealing Temperature (Tm), and ΔG of primer self-folding, and identify covered species and functional groups. Using this new platform, we designed primers for several functional genes in the nitrogen cycle, including *napA* and *amoA*. Our newly designed primers achieved higher coverage than the commonly used primers for all tested genes. The program and the associated platform that applied the *k*-mer algorithm could greatly enhance the design and evaluation of primers for environmental microbiome studies.

## 1. Introduction

At present, studies of environmental microbial community composition and diversity generally follow the procedure of sample collection, DNA extraction, Polymerase Chain Reaction (PCR) amplification, and high-throughput sequencing [1]. PCR amplification and high-throughput sequencing have greatly changed our understanding of microbial diversity in various environments by bypassing the need for cultivation [2]. Amplicon sequencing is a highly targeted method for analyzing gene or species variations across broad taxonomic ranges [3], using primers to amplify specific regions within environmental DNA to enrich the targeted sequences [4]. Thereafter, high-throughput sequencing is carried out to capture the genetic diversity of the amplified products. Since PCR is the central step of this procedure, the design of primers to amplify the target gene is the key step to determine the specificity and coverage of amplicon sequencing for environmental microbiome studies. The amplification primers of marker genes, such as the commonly used primers for prokaryotic 16S rRNA gene and fungal Internal Transcribed Spacer (ITS) region, which are often used in diversity surveys, have been widely used and evaluated [5,6]. However, with recent expansions in sequencing capabilities due to next-generation sequencing, the existing primers for other functional genes are unable to cover broad taxonomic groups with their variable sequences. Only by designing suitable primers can PCR reactions for the detection of taxonomic and functional genes be carried out accurately.

Coverage and specificity are both extremely important issues for the quality of primers. The coverage of a primer, that is, the proportion of sequences within a given sequence set that is matched, can be improved by introducing nucleic degeneracy, however, higher nucleic degeneracies in a primer might negatively affect its specificity. Thus, balancing the nucleic degeneracy is also important for the primer design in microbiome studies. In addition, current primer design methods are mainly based on aligned DNA sequences. The most highly conserved region of a target gene generally has a similar sequence across many different species. Thus, high-coverage primers could be designed in these conserved regions [7]. Many software programs for the design of degenerate primers are based on this pattern, including HYDEN [7], PrimerProspector [8], and DegePrime [9]. However, with the rapidly increasing number of sequences deposited in public databases, the time cost of multiple sequence alignment has vastly increased. With high requirements for sequence length and quality, the accurate alignment of thousands of sequences may take several days, while tens of thousands of sequences may run for dozens of days [10]. Therefore, there is an acute need for a new, fast, and accurate method to design primers for amplicon sequencing technology, especially for targeting microbial functional genes.

The *k*-mer algorithm is a commonly used algorithm to improve assembly effect in the analysis of metagenomic sequencing results [11]. It divides the sequence into a string containing *k* bases. Generally, a sequence with the length of m can be divided into *m* − *k* + 1 *k*-mers. Using the sequence AACTGACTGA as an example: When *k* = 4, for the input sequence, we start from the first base and use the form of sliding window (step size is 1) to extract the 4 bp sequence successively. This sequence can be divided into 7 *k*-mers: AACT, ACTG, CTGA, TGAC, GACT, ACTG, and CTGA. At present, although the *k*-mer algorithm is widely used in bioinformatics, such as genome assembly, genome sequencing coverage evaluation, error correction of sequencing data, multiple sequence alignment, and repeat sequence detection, it has not as yet been used in the development of primer design methods, especially for the design of degenerate primers. Because the conserved region of a gene should be the same or very similar across multiple sequences, we infer that the frequent *k*-mers should originate from those regions and be detectable when the *k* value is greater than a certain length. In view of this, we provided a new primer design method based on the *k*-mer algorithm, which can avoid the tedious step of sequence alignment and greatly improve the running speed while still ensuring accuracy. This paper provided a theoretical guidance for a new idea of microbial functional gene amplification primer design. Based on this algorithm, we designed degenerate primers for several microbial functional genes, and verified their coverage and specificity.

## 2. Materials and Methods

### 2.1. Analysis Platform of ARDEP

We developed a bioinformatic platform for our rapid degenerate primer design program (ARDEP, available at http://mem.rcees.ac.cn:8082/) for amplicon microbiome studies, to which users may upload sequence databases for primer design and assessment. The analysis platform allows for sequence length statistics, length filtering, and redundancy removal to be performed on the sequence database using tools located in the “Sequence Processing” section. Within this section, “Length statistics” provides quick information about sequence length distribution. “Trim by Sequence Length” is a program to trim sequences based on length. Only sequences longer than (or equal to) the minimum length and shorter than (or equal to) the maximum length are retained. Users are able to choose between either deleting the longer sequences or trimming them to the required length. “Remove same sequences for FASTA” is for the removal of same exact sequences from the database. After performing sequence database quality control, users may run ARDEP for primer design in the “Primer Processing” section. The tools within this section also allow users to evaluate primers. “Coverage Calculator” could calculate primer coverage using Nucleotide Basic Local Alignment Search Tool (BLASTn). “Amplification Product Length” allows for calculation of amplification product length. If the uploaded database contains taxonomic information, “Covered Taxonomy Calculator” can be used to count the sequence number of taxonomy and functional group covered by primer sets. This tool not only counts the covered sequence number of taxonomy, but also assesses covered types of gram-positive/negative and oxygen requirement as based on the BugBase database [12].

### 2.2. Algorithm of ARDEP

#### 2.2.1. Division of Database Sequences by *k*-mer Program 

Based on the *k*-mer algorithm, primer length is set as *k*, and all sequences are divided into *k*-mers (Figure 1). If *k* is set as a length range (k_1_ < *k* < k_1+n_), then *n* times will be calculated according to different *k* values. The *k*-mer statistics are calculated using jellyfish software [13].

#### 2.2.2. Degenerate Primer Combination

All *k*-mers with overlap > *x* will be combined because *k*-mers with overlap > *x* are defaulted to exist in the position of primer length-*x* base difference. The value *x* is set by the user, with the default value set as half of the primer length. The *k*-mer with the highest frequency is retained. If the frequency of two *k*-mers’ overlap > *x* are the same, the longer *k*-mer will be kept. For example, the frequency of one *k*-mer CGCGGTTGCTCGTGTGG is 1218, and the frequency of the other k-mer GTTGCTCGTGGCGCG is 1137. By default, these two *k*-mers are located four bases apart. Therefore, only CGCGGTTGCTCGTGTGTGG will be retained. According to how frequently a *k*-mer appears, potentially dozens of high-frequency *k*-mers could be selected for downstream processing. The *k*-mers which contain 1 base different from the high-frequency *k*-mers are combined with the high-frequency *k*-mer in the form of a degenerate base. The preliminary primers have been designed. For example, searching for *k*-mers of coverage >1% and with a base difference between the high frequency *k*-mer ACATGCCAATGCTGGT, we find two *k*-mers, ACATGCGATGCTGGT and ATATGCCAATGCTGGT. These can be combined in the form of degenerate bases to AYATGCCRATGCTGTGT (Y: C or T, R: A or G). According to the amplification product fragment length and coverage, the most suitable primer sets for the database are selected.

#### 2.2.3. Calculation of Basic Primer Properties

The annealing temperature (Tm), secondary structure (ΔG of primer’s self-folding), and GC content (GC%) of primers are calculated as follows:GC% = (G + C)/primer lengthAnnealing temperature (Tm): when the primer length is shorter than 20 bp: Tm = 4 (G + C) + 2 (A + T); when the primer length is longer than 20 bp, Tm = 62.3 °C + 0.41 °C (GC%) − 500/primer length.The stability parameter ΔG, which is the free energy needed for DNA double-strand formation, reflects the stability of the double-strand structure. If the complementary base of the primer is too long, a primer dimer or hairpin structure could be produced, which leads to high ΔG and the PCR reaction cannot proceed normally. This parameter is calculated by *mfold* software [14].

#### 2.2.4. Coverage Verification and Covered Taxonomy/Group Calculation

The primer coverage and amplification product fragment length are calculated, and the main groups that the primer set could cover are detected, including covered taxonomy, oxygen requirement type, and gram type (i.e., gram-positive and gram-negative). The classifications of oxygen requirement and gram type are based on the species phenotype database of BugBase [12].

#### 2.2.5. Parameters Required by ARDEP Platform

Number of database sequences;Primer length (maximum and minimum, default 18–20 bp);Amplification product fragment length (maximum and minimum, default value is 200–500 bp);Minimum coverage users could accept of the output primer (the default value is 20%, i.e., 0.2);Overlap of high-frequency *k*-mers to be merged (the default value is 10);DNA folding temperature in PCR experiment (the default value is 57 °C);Ionic conditions of Mg++ (the default value is 20) and Na+ (the default value is 0) in PCR experiment (Figure 2).

### 2.3. Test Datasets

We chose the denitrification gene *napA* during the nitrogen cycle to test the ARDEP primer design tool. The *napA* gene could encode proteins that catalyze the reduction of NO_3_^−^ to NO_2_^−^ (periplasmic-bound nitrate reductase). In order to design reliable primers for the *napA* gene, the marker gene of dissimilatory nitrate reductase in the dissimilatory nitrogen reduction to ammonia (DNRA) process, we collected a credible *napA* functional gene database. Based on 472 *napA* sequences that were retrieved from GeoChip probe database [15], we screened for sequences with lengths of 500–1000 bp. These sequences were aligned using the Muscle program [10] and their similarity was calculated. Only the sequences with similarity >80% were retained and duplicate sequences were cut such that only one sequence per species was kept. Finally, 126 highly conserved sequences were chosen as seed sequences. Using HMMER, we searched for the conserved domain of homologous proteins in the GenBank Protein Database [16,17], gathering all homologous sequences which could be annotated as *napA*. Redundant sequences and unqualified sequences were removed, resulting in 38,768 sequences being retained. According to the *E*-value in HMMER (*E* < 10^−5^) and the annotation information of sequences (must include at least one set of “periplasic nitrate reduce, *napA* gene product, *napA*, nitrate reduce, nitrate oxidoreductase” as key words), we selected 3397 highly credible sequences from the collected homologous sequences. On this basis, we downloaded the corresponding protein and nucleic sequences according to the protein accession number. Through the taxid corresponding to the protein accession number, we supplemented the taxonomy information to complete the database construction. These sequences and their annotation were used to test the ARDEP primers design tool. In order to calculate the specificity of the designed primers for the *napA* gene, we constructed the *narG* gene database which also reduced NO_3_^−^ to NO_2_^−^ (membrane-bound nitrate reductase). The construction of the *narG* sequence database was based on 47 seed sequences. Similar to *napA* we searched for the conserved domain of homologous proteins in the GenBank Protein Database through HMMER [16], gathering all homologous sequences which may be annotated as *narG* (*E* < 10^−5^ while including at least one set of “nitrate reductase, nitrate-reductase, dissimilatory membrane-bound nitrate reductase, *narG* gene product, nitrate reductase alpha subunit, respiratory nitrate reductase, *narG*, molybdopterin” as key words). Thus, 30,217 sequences were retained after removal of redundant and unqualified sequences. On the other hand, the specificity was also evaluated based on the NCBI nt database (58,992,497 sequences, updated date: 6 June 2020) to rule out the possibility of primers overwriting other homologous or similar genes.

To test the applicability of our method to a relatively large database, we downloaded 119,096 sequences from NCBI and 323,085 sequences from FunGene [18] that were annotated as *amoA.* FunGene is a free database containing redundant sequences of a number of functional genes of microorganisms. These *amoA* sequences were merged to remove redundant sequences, and 173,790 sequences with length of 200–1000 bp were retained. On the other hand, we built ammonia-oxidizing archaea (AOA) and ammonia-oxidizing bacteria (AOB) databases based on 52 seed sequences through HMMER [16]. All homologous sequences were gathered which may be annotated as *amoA* (*E* < 10^−5^ while including at least one set of “ammonia monooxygenase, ammonia monooxygenase subunit A, *amoA* gene product, putative ammonia monooxygenase subunit A, *amoA*” as key words). After removing redundancy, these sequences of 350–650 bp length were divided into AOA and AOB according to their taxonomy annotation; 45,771 AOA sequences and 42,659 AOB sequences were used to design AOA and AOB primers by ARDEP, respectively. In order to calculate the specificity of the designed primers for the *amoA* gene, a total of 22,357 nonredundant *pmoA* sequences were collected from NCBI, including 1165 archaea sequences and 21,192 bacteria sequences. The *pmoA* gene is a homologous gene of *amoA*, which is, however, involved in completely different ecological processes. On the other hand, the specificity was also evaluated based on the NCBI nt database to rule out the possibility of primers overwriting other homologous or similar genes.

## 3. Results

### 3.1. Primer Design of Functional Gene napA

The *napA* sequence database included 3397 nonredundant *napA* gene sequences of high credibility and identified species. These sequences were mainly composed of *Proteobacteria*, accounting for 71.65% of the sequences, while the remaining were 14.72% *Firmicutes*, 12.42% *Actinobacteria*, and 1.21% unclassified phyla. Based on BugBase classification [12], 52.46% of the sequences were classified as facultative anaerobic, 33.41% were classified as aerobic, and 2.24% were classified as anaerobic. According to the gram classification, 71.77% of the sequences were classified as gram-negative, while 26.58% were classified as gram-positive bacteria. At present, the most widely used primers for the *napA* gene are v16cf-GCNCCNTGYMGNTTYTGYGG/v17cr-RTGYTGRTTRANANCCCATTCCA, which could only cover less than 1% of the sequences in this database, with an amplicon length of 1040 bp. In order to design primers with higher coverage and an assignable amplification product fragment length for high-throughput sequencing, we used ARDEP to design new primers based on our *napA* sequence database. Through the *k*-mer algorithm, all sequences in the database were cut into *k*-mers with the lengths from 19 to 20 bp. The basic properties of primers were calculated after combining degenerate bases. The ARDEP output file of individual primers and their basic properties is shown in Appendix A. The primer named as napA_kmer5 (TTYTAYGACTGGTAYKSYGA) possessed the highest coverage, covering 80.01% of all sequences. The top three primers with the highest coverage and their basic properties are listed in Table 1.

Based on the results of the preliminary design, primer sets were recommended according to the amplification product fragment length. An ARDEP output file of paired primers is shown in Appendix A. According to the *napA* gene length and sequencing feasibility, we determined the optimal product length as 200–500 bp. Compared with the most used primer set v16cf/v17cr, the coverage of the *napA* gene was highly increased and the amplicon length was more optimal for high-throughput sequencing. Primer set napA_kmer5/napA_kmer8 with the highest coverage of 65.50% could amplify product with length 421 bp. Another primer set, napA_kmer6/napA_kmer8, with the coverage of 64.20% could amplify product with length 466 bp (Table 1). Tm of forward and reverse primers of both primer sets was similar, which meant they were suitable to use in pairs. We carried out further assessment of these two primer sets and calculated their covered species, oxygen requirement, and gram type (Figure 3). Primer set napA_kmer5/napA_kmer8 could cover 87.59% *Proteobacteria* sequences, 18.00% *Actinobacteria* sequences, and 0.71% *Firmicutes* sequences (Figure 3a). Although the coverage of napA_kmer6/napA_kmer8 was a little lower than napA_kmer5/napA_kmer8, this primer set could cover more *Proteobacteria* sequences (89.44%). However, it could barely cover *Actinobacteria* and *Firmicutes* sequences (Figure 3a). For oxygen demand types, napA_kmer5/napA_kmer8 could cover more aerobic sequences, while napA_kmer6/napA_kmer8 could cover more anaerobic sequences (Figure 3b). For gram types, both primer sets could cover more than 85% gram-negative sequences. However, napA_kmer5/napA_kmer8 could cover 10.30% gram-positive sequences, while napA_kmer6/napA_kmer8 could barely cover any (Figure 3c). These two primer sets showed different coverage bias to our database.

The specificity of these primers was evaluated based on the NCBI nt database and *narG* database that we created (Table 1). The coverage of all these *napA* primers for the nt database was less than 0.01%. In other words, these primers have quite good specificity to the *napA gene.* For the *narG* database, the coverage of all designed primers was less than 0.82% (Appendix A). In other words, our primers have good specificity even for homologous genes with similar sequences. The entire primer design process took a total of 13 min. In the same computing environment, the computing time required for DegePrime, for primers designed with the highest coverage of 63.97% (TTYTAYGACTGGTAYTGCGA), was 1 h and 41 min. The calculation time included MAFFT [19] sequence alignment calculation with maximum parallelism. Therefore, our method not only designed faster, but produced primers with higher coverage as well.

### 3.2. Primer Design of Functional Gene amoA

The most commonly used forward primer for AOA, Arch-amoA-for (CTGAYTGGGYTGGACATC), could cover 44.18% of AOA sequences in our database, while the reverse primer Arch-amoA-rev (TTCTTCTTTGTTGCCCAGATA) could cover 46.76% of AOA sequences. As a set, these primers could cover 41.17% of AOA sequences. The common forward primer for AOB, amoA_1F (GGGGHTTYTACTGGTGGT), could cover 50.07% of AOB sequences, while the reverse primer amoA_2R (CCCCTCKGSAAAGCCTTCTTC) could cover 54.77% of AOB sequences. Together as a set, the primers could only cover 43.97% of AOB sequences. We designed new primers for both AOA and AOB *amoA* sequence databases using ARDEP to see if we could get primers with higher coverage.

For 45,771 AOA sequences, we designed nine degenerate primer sets with coverage of over 50% (Table 2). The primer length was 19–20 bp and the amplification product fragment length was 200–500 bp. The coverage and basic properties of the individual primers were also output in the results (Appendix A). AOA_kmer14 was the primer with the highest coverage at 88.99% of sequences, while AOA_kmer9 also reached a high coverage of 86.29%. In addition, AOA_kmer4, AOA_kmer2, and AOA_kmer5 could cover more than 70% AOA sequences. The output file of AOA primer sets is shown in Appendix A. The primer set AOA_kmer4/AOA_kmer14 showed the highest coverage at 66.07%. This primer set could amplify products of 427 bp length. We evaluated the specificity of these designed AOA primers using the *pomA* database we constructed. The coverage of all designed AOA primers to the *pmoA* database was <0.01%. In other words, our AOA primers can hardly cover its homologous gene *pmoA*. The specificity of these AOA primers was also evaluated based on the NCBI nt database to see if they could cover other functional genes. The coverage of these primers for the nt database was less than 0.06% (Table 2). Compared with AOA database sequence number and primer coverage, we concluded that the specificity of these primers was reasonable for the archaea *amoA* gene. The entire design process of ARDEP took 8 h and 3 min.

For the 42,659 AOB sequences, we designed nine degenerate primer sets with coverage over 50% (Table 3). The primer length was 19–20 bp and the amplification product fragment length was 200–500 bp. AOB_kmer6 was the primer with the highest coverage at 84.12% of AOB sequences (Appendix A). The output file of AOA primer sets is shown in Appendix A. The primer set AOB_kmer3/AOB_kmer6 had the highest coverage at 55.50%. We evaluated the specificity of these designed AOB primers using the *pomA* database we constructed. The coverage of all designed AOB primers to the *pmoA* database was less than 0.06%. The specificity of these AOB primers was also evaluated based on the NCBI nt database. The coverage of these primers for the nt database was less than 0.05%. Compared with AOB database sequence number and primer coverage, we concluded that the specificity of the designed primers was reasonable for the bacterial *amoA* gene. The entire design process took 7 h and 47 min.

For all 173,790 nonredundant *amoA* sequences, including both AOA and AOB, we designed three degenerate primer sets with coverage of over 20% (Table 4). The primer length was 18–20 bp, and amplification product fragment length was 200–500 bp. Among these primers, *amoA*_kmer15 and *amoA*_kmer20 could, individually, reach a coverage of more than 36%. As AOA and AOB are often studied separately, the primer coverage designed based on the mixed database of the *amoA* gene is relatively low. The specificity of these *amoA* primers was evaluated based on the NCBI nt database. The coverage of these primers for the nt database was less than 0.1% (Table 4). amoA_kmer7 could cover the most sequences in the nt database with the coverage of 0.06%, namely 35,396 sequences. Compared with *amoA* database sequence number and primer coverage, we concluded that the specificity of the designed primers was reasonable for the *amoA* gene. For the homologous gene *pmoA* which we pay special attention to, some primers showed certain preference. For example, the ability of amoA_kmer20 to cover the *amoA* gene is basically the same as *pmoA*. For the primer with poor specificity, it could be used with another primer with better specificity as well. For example, when paired with amoA_kmer6, the specificity of this primer set had become better. In this way, the amplification efficiency can also be guaranteed. The entire design process took 12 h and 55 min.

## 4. Discussion

With the development of molecular biology technology and metagenomic technology, amplicon sequencing has greatly changed our understanding of microbial diversity. PCR is one of the most basic and important tools in these technologies, and thus primer design and assessment have become key steps to ensure the accuracy of environmental detections [20,21]. The conventional programs for primer design across broad taxonomic ranges have all been based on aligned nucleic sequence [8,9]. However, the process of sequence alignment is highly time-consuming, with the alignment of thousands of sequences often taking dozens of hours. We have tested several sequence alignment programs in routine experiments and found that: (i) when more than 2000 *napA* sequences were aligned, the calculation time of MAFFT was 33 h and 23 min with less than 10 parallelism, and 45 min with maximum parallelism; (ii) when more than 4000 *napA* sequences were aligned, the calculation time of Clustal Omega [22] was 1 h and 33 min, while MAFFT, with high accuracy, took more than 1 h with maximum parallelism; (iii) when more than 10,000 *nirA* gene sequences were aligned, the high-accuracy program MUSCLE took 88 h and 7 min (i.e., more than three days) while the fastest method Clustal Omega required 3 h and 5 min for the necessary calculations; (iv) when more than 30,000 *amoA* gene sequences were compared, MUSCLE required 156 h and 22 min (i.e., more than six days) while Clustal Omega took 4 h and 20 min for calculations (Appendix A). These slow speeds greatly increase the time cost of primer design, and inaccurate alignments would likely decrease the possibility of successful primer searches in the conserved regions.

In order to shorten the calculation time of primer design, bypassing the sequence alignment seemed to be a logical route. The *k*-mer algorithm is often used to improve the assembly effect in the analysis of metagenomic sequencing results, but has never been used in primer design before. Theoretically, the conserved regions can be located for use as primers according to the frequency of *k*-mers without the need for alignment. Based on this, we designed a method for primer design with greatly improved calculation speed. Our results showed that the database calculation time of more than 3000 sequences was only about 13 min and for a database of more than 40,000 sequences, it took about 8 h each for AOA and AOB primer design. The calculation time for a database of over 170,000 sequences was 12 h and 55 min. Referring to the results of test data, we have created some rough statistics on the calculation time and data size. The calculation time included *k*-mer sequence division, degenerate primer merging, amplification product fragment length calculation, primer basic property calculation, covered group calculation, reading and writing files, and statistical calculations. There can be differences in the calculation time for the same sized dataset as calculation time is related to the degeneracy and sequence similarity.

The design and assessment of primers are closely related to how reliable the sequence database is. Therefore, choosing a credible and complete database is very important for primer design. For the *napA* gene database in our study, we used the method of protein conserved domain alignment to construct a database based on 472 *napA* sequences that were retrieved from the GeoChip probe database. According to the E-value in HMMER and the annotation information of sequences, we selected 3397 highly credible sequences from the collected homologous sequences from NCBI. These sequences are with high reliability and complete description information. For the functional genes of environmental microorganisms, numbers of sequence databases have already been completely built. In addition to the synthetic databases FunGene and Genbank mentioned in this article, there are many targeted databases, such as antibiotic resistance gene database SDARG [21], nitrogen cycling gene database NCycDB [23], et al. These targeted databases mostly carried out taxonomy annotation and reliability classification of sequences, which are also suitable for primer design and assessment. Our analysis platform could perform secondary screening on the length, redundancy, and other aspects of these database sequences. With high reliability of sequence database, the designed primers could exhibit good performance in real experiments. For AOB, we have performed an experiment to detect the diversity of soil AOB by *amoA* amplicons using designed AOB primer set AOB_kmer2/AOB_kmer28 (Appendix A). From our preliminary amplification results (Appendix A), this designed primer set had high specificity in the tested soil samples, showing only one clear band in the proposed amplicon length (~250 bp).

The advantage of ARDEP is not only reflected in the calculation speed. Compared with other primer design software, it provides more information. ARDEP allows users to provide a length range to design primers, rather than limiting them to a fixed length. ARDEP could also calculate the GC%, Tm, and ΔG of primers as a reference for experiments. It also provides recommendations for combinations of primer sets according to coverage and amplification product length. Meanwhile, most primer design software based on sequence alignments only provide the location of a single primer [9]. In addition, ARDEP could calculate the organisms covered by primer sets according to taxonomic annotations. This tool allows the user to be more specific in primer selection according to the studied environment. Our method provided theoretical guidance for the design and selection of microbial functional gene primers. However, the specific expression of primers in amplification still needs to be explained by PCR experiments and sequencing results. In general, with the ever-increasing amount of data that is available, our method has a strong time cost advantage, and provides a new approach for primer design and sequence analysis.

## Figures and Tables

**Figure 1 ijerph-17-05958-f001:**
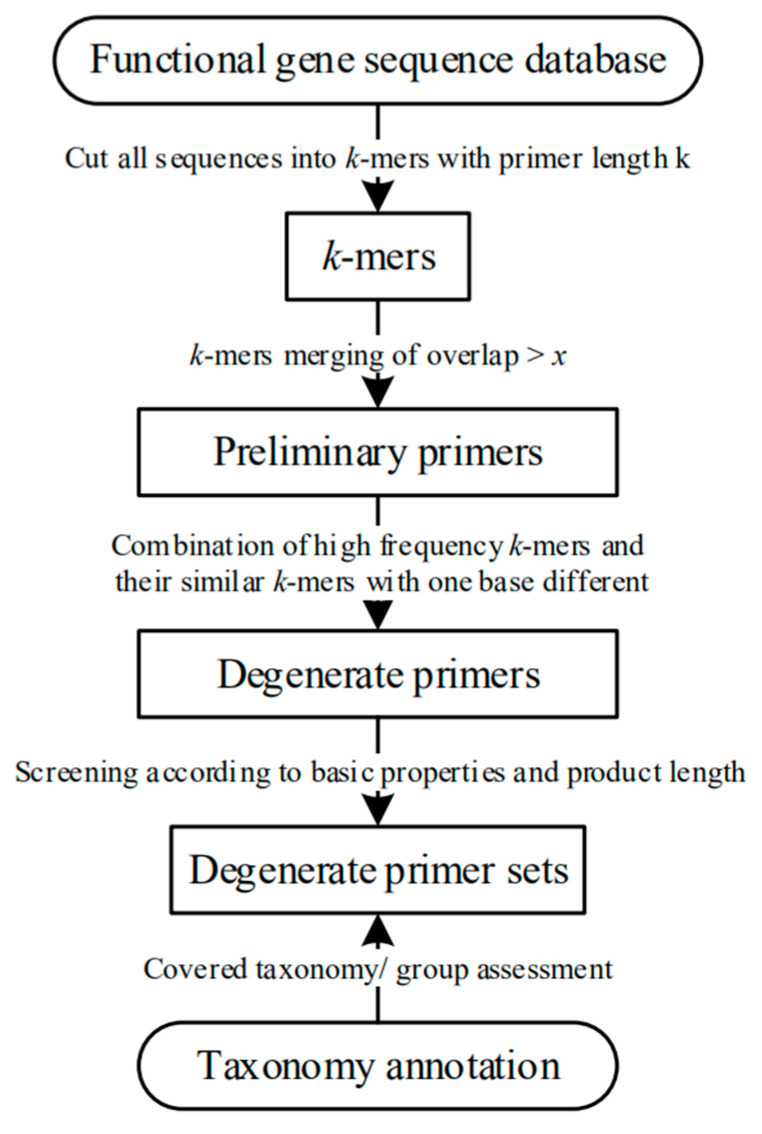
Workflow for primer design process.

**Figure 2 ijerph-17-05958-f002:**
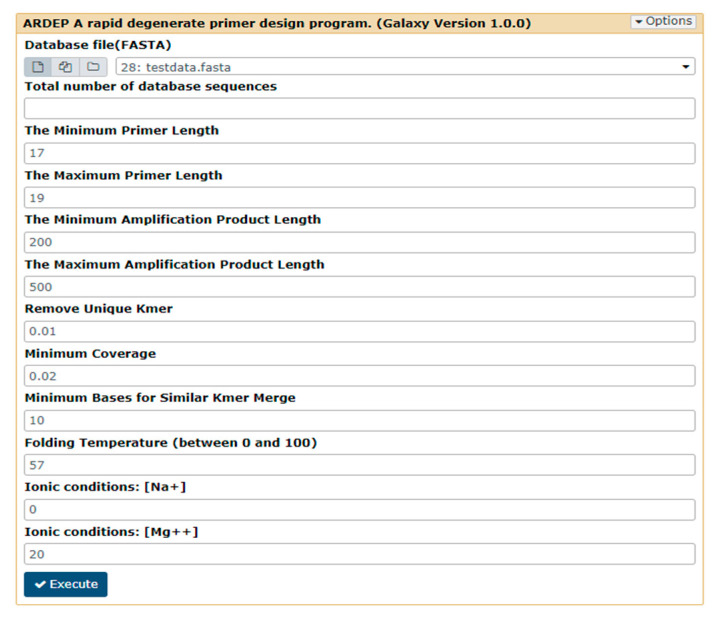
The *k*-mer-based primer design tool interface. The default values for program operation parameters are shown in the figure.

**Figure 3 ijerph-17-05958-f003:**
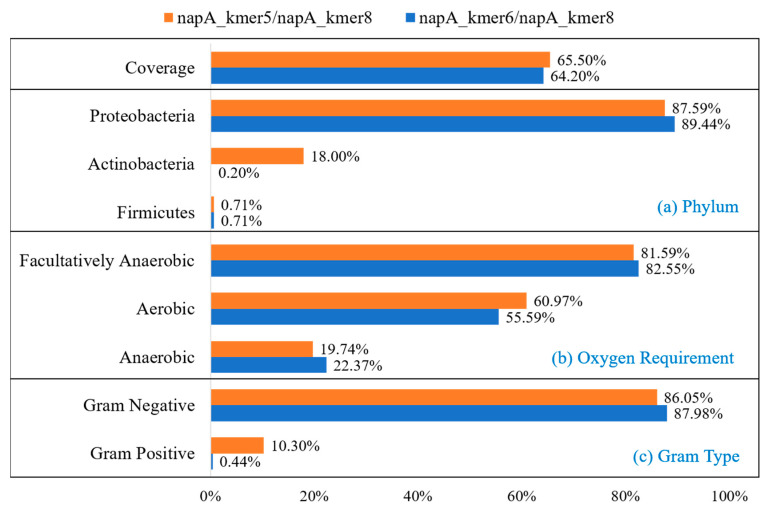
Covered sequence composition of two *napA* primer sets. (**a**) Phylum; (**b**) oxygen requirement type; (**c**) gram type.

**Table 1 ijerph-17-05958-t001:** *napA* primers with basic properties.

Primer Name	Primer Sequence	Product Length	Coverage		Specificity		Primer Properties		
					narG	nt	ΔG	GC%	Tm
napA_kmer5	TTYTAYGACTGGTAYKSYGA	421	80.01%	65.50%	0.82%	<0.01%	(−2.07)–(1.70)	30.00–55.00%	52.00–62.00
napA_kmer8	ACSTGGGADAYNCADCADAC		67.91%				(−3.15)–(2.10)	40.00–65.00%	56.00–66.00
napA_kmer6	TGGGGYGAVCADACNGAYGT	466	77.77%	64.20%	0.80%	<0.01%	(−1.65)–(1.24)	45.00–70.00%	58.00–68.00
napA_kmer8	ACSTGGGADAYNCADCADAC		67.91%				(−3.15)–(2.10)	40.00–65.00%	56.00–66.00

**Table 2 ijerph-17-05958-t002:** Recommended primer sets for *amoA* gene of AOA.

Primer		Coverage	Product Length	Specificity	
Forward	Reverse		Mode	*pmoA*	nt
AOA_kmer4	AOA_kmer14	66.07%	427	<0.01%	<0.01%
AOA_kmer2	AOA_kmer9	62.90%	202	<0.01%	0.06%
AOA_kmer2	AOA_kmer5	56.08%	433	<0.01%	0.06%
AOA_kmer4	AOA_kmer5	55.65%	385	<0.01%	0.06%
AOA_kmer8	AOA_kmer14	55.18%	472	<0.01%	<0.01%
AOA_kmer2	AOA_kmer15	52.71%	407	<0.01%	<0.01%
AOA_kmer4	AOA_kmer6	52.14%	211	<0.01%	0.05%
AOA_kmer2	AOA_kmer11	52.09%	229	<0.01%	<0.01%
AOA_kmer4	AOA_kmer15	51.35%	359	<0.01%	<0.01%

**Table 3 ijerph-17-05958-t003:** Recommended primer sets for *amoA* gene of AOB.

Primer		Coverage	Product Length	Specificity	
Forward	Reverse		Mode	*pmoA*	nt
AOB_kmer3	AOB_kmer6	55.50%	204	0.06%	0.05%
AOB_kmer1	AOB_kmer22	53.66%	439	<0.01%	<0.01%
AOB_kmer6	AOB_kmer26	51.83%	218	<0.01%	<0.01%
AOB_kmer3	AOB_kmer17	51.63%	204	0.06%	<0.01%
AOB_kmer1	AOB_kmer5	51.30%	448	<0.01%	0.04%
AOB_kmer6	AOB_kmer22	51.12%	294	<0.01%	<0.01%
AOB_kmer4	AOB_kmer6	50.87%	228	<0.01%	0.04%
AOB_kmer1	AOB_kmer27	50.79%	451	<0.01%	<0.01%
AOB_kmer17	AOB_kmer26	50.10%	218	<0.01%	<0.01%

**Table 4 ijerph-17-05958-t004:** Recommended primers with basic properties for *amoA*.

Primer	Sequence	Coverage		Specificity				DeltaG	GC%	Tm	Product Length (Mode)
				pmoA		nt					
amoA_kmer7	ATYAAYGCAGGRGACTAYAT	27.62%	22.90%	<0.01%	<0.01%	0.06%	<0.01%	(−1.65)–(−0.74)	30.00–50.00%	52.00–60.00	427
amoA_kmer15	AADTTCTAYAAYAGYCCHG	36.45%		<0.01%		<0.01%		(−0.03)–(1.29)	26.32–52.63%	48.00–58.00	
amoA_kmer6	GTVTGGTGGTAYYTTGGYAA	22.40%	20.95%	0.17%	0.03%	0.05%	<0.01%	(−0.49)–(1.86)	35.00–55.00%	54.00–62.00	202
amoA_kmer20	TYTAYCCNGGYAACTGGMC	36.82%		35.83%		<0.01%		(−4.91)–(0.43)	42.11–68.42%	54.00–64.00	
amoA_kmer2	GGTTTCTACTGGTGGTCVCA	23.93%	20.28%	<0.01%	<0.01%	0.05%	<0.01%	(−0.15)–(0.40)	50.00–55.00%	60.00–62.00	448
amoA_kmer17	GAAGAAGGCTTTSCMGAGG	22.84%		<0.01%		<0.01%		(−0.66)–(−0.47)	52.63–57.89%	58.00–60.00	

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
