# Peer review of "ARDEP, a Rapid Degenerate Primer Design Pipeline Based on k-mers for Amplicon Microbiome Studies"

_ijerph, 2020, doi:10.3390/ijerph17165958_

Round 1

Reviewer 1 Report

The manuscript entitled “ARDEP, A rapid degenerate primer design pipeline based on k-mers for amplicon microbiome studies” developed a new primer design method which bypassed the time-consuming alignment step through using a k-mer based algorithm, this method dramatically reduced the running time while still kept the accuracy. Meanwhile the authors developed an open access platform, beside the primer design function, with this platform the users could calculate the PCR product length, GC content, Tm et.al. And the system was evaluated with a number of N-cycling genes.

The topic of the paper fits the journal. Considering the huge demand for design primers based on the daily increasing gene database, the paper is attractive for a very wide range of readers. Therefore I would suggest a publication of this paper if the author could consider the following two points.

  1. Degenerate primers are widely used in environmental sample studies, it increase the coverage of the primer and can capture the same gene with slightly different sequences, however if the primer is made of too many degenerate bases, the specificity declines, did the author consider how to balance between the coverage and specificity?
  2. In the study, the primer design for N cycle genes was performed with Genbank and Fungene, however both of these databases were not curated, what is the advantage for using these databases?

Author Response

Thank you very much for your summary and affirmation of our research! We have made changes to the article in response to your two comments. Below listed the point-to-point responses to your feedbacks.

  1. Degenerate primers are widely used in environmental sample studies, it increase the coverage of the primer and can capture the same gene with slightly different sequences, however if the primer is made of too many degenerate bases, the specificity declines, did the author consider how to balance between the coverage and specificity?

Yes, the degenerate primers are widely used in the study of environmental samples, especially primers for functional genes of microorganisms. Although the sequences of these functional genes are similar across species, slight differences still exist. The addition of degenerate bases could improve the primer coverage. However, too many degenerate bases could reduce the specificity of primers. Therefore, we considered the coverage of each degenerate primer to the database to control the amount of degeneracy bases (section 2.2.2 in our manuscript). In our current algorithm, degenerate primers with coverage of more than 1% will merge with main k-mers (highest coverage). This is a relatively reasonable value for our results so far. Our results also indicated that the degeneracy primers designed by ARDEP have good specificity for both homologous gene databases and nt databases (Table 1-4). For example, the degeneracy of AOB_kmer22 (GCNTWYGGYGAAGAAGGC) is 32. This primer could cover 60.08% of the sequences in the AOB database. Its specificity to both pmoA and NT databases was less than 0.01%.

  1. In the study, the primer design for N cycle genes was performed with Genbank and Fungene, however both of these databases were not curated, what is the advantage for using these databases?

Response: Thank you for your comment on database selection. The selection of database is the basis of primer design. We believe that no matter which method is used to design primers, the more comprehensive the database is, the more sequences could be covered by the designed primers. If the database sequence can reduce the composition of microbial community structure, the sequencing results will better reflect the real situation of environmental microorganisms. For napA gene database, we used the method of protein conserved domain alignment (HMMER) based on 472 napA sequences that were retrieved from GeoChip probe database. According to the E-value in HMMER (E < 10-5) and the annotation information of sequences, we selected 3397 highly credible sequences from the collected homologous sequences from NCBI. These sequences are with high reliability and complete description information. In order to test the applicability and calculating speed of our method to a relatively large database, we downloaded 119,096 sequences from NCBI and 323,085 sequences from FunGene that were annotated as amoA and merge them together. Because these sequences are not well curated, the coverage of our designed primers is not very high against the database. Therefore, we described interpretations of the database selection in the discussion section and added more description of the advantages of our database.

(Page 12, Line 346) “The design and assessment of primers are closely related to how reliable the sequence database is. Therefore, choosing a credible and complete database is very important for primer design. For napA gene database in our study, we used the method of protein conserved domain alignment to construct a database based on 472 napA sequences that were retrieved from GeoChip probe database. According to the E-value in HMMER and the annotation information of sequences, we selected 3397 highly credible sequences from the collected homologous sequences from NCBI. These sequences are with high reliability and complete description information. For the functional genes of environmental microorganisms, numbers of sequence databases have already been completely built. In addition to the synthetic databases FunGene and Genbank mentioned in this article, there are also many targeted databases, such as antibiotic resistance gene database SDARG [21], nitrogen cycling gene database NCycDB [23], et al. These targeted databases mostly carried out taxonomy annotation and reliability classification of sequences, which are also suitable for primer design and assessment. Our analysis platform could perform secondary screening on the length, redundancy and other aspects of these database sequences. ”

Reviewer 2 Report

Wu et al. proposed a new primer design method, which can eliminate sequence alignment. It is based on the KMER algorithm commonly used in metagenomic assembly. An analytical website for sequence processing and primer design is also proposed in this paper. The authors used this method to design new primers for several functional genes. Coverage, specificity and other properties of the primers were evaluated. It has been greatly optimized in terms of running time as well. This study is innovative and applicable to the study of environmental microorganisms, and the results are suitable for publishing in IJERPH with appropriate revisions. Specific suggestions for revisions are as follows.

  1. Line 62 Using the sequence aactgactga as an example:

The sequence aactgactga could be represented in capital letters.

  1. Line 109 these two k-mers are located in the same position, with a difference of four bases.

These two k-mers are four bases apart. So “located in the same position” is not accurate.

  1. Line 315-324

This part of the discussion is all about describing the running time. I suggest putting them into a table that will be easier to read.

  1. Line 346

The discussion in this section introduced the selection of targeted databases. However, the reliability and advantages of the four databases in this paper have not described in discussion section. I suggest adding some description of the advantages for the database you choose.

  1. Line 335 Referring to the results of test data, we have created some rough statistics on the calculation time and data size (Table 6).

The description did not correspond with the table.

Author Response

Response: Thank you very much for your summary and all your below comments about the details. We have made changes to the article in response to your two comments. Below listed the point-to-point responses to your feedbacks.

1.Line 62 Using the sequence aactgactga as an example:

The sequence aactgactga could be represented in capital letters.

Response: Thanks for your correction, the sequence in this article has been changed to uppercase

2.Line 109 these two k-mers are located in the same position, with a difference of four bases.

These two k-mers are four bases apart. So “located in the same position” is not accurate.

Response: Thank you for your correction. The sentence in the passage has been modified to “By default, these two k-mers are located four bases apart.”

3.Line 315-324

This part of the discussion is all about describing the running time. I suggest putting them into a table that will be easier to read.

Response: Thank you for your suggestion. We have added Table S7 to the supplementary material to present the comparison of operation times more clearly.

Sequence Number

Gene Name

Sequence Alignment Method

Sequence Alignment Calculating Time

ARDEP Calculating Time

2000

napA

MAFFT

(with maximum parallelism)

45 minutes

~10 minutes

4000

napA

Clustal Omega

1 hour and 33 minutes

~15 minutes

MAFFT

(with maximum parallelism)

1 hour

>10000

nirA

MUSCLE

88 hours and 7 minutes

Clustal Omega

3 hours and 5 minutes

>30000

amoA

MUSCLE

156 hours and 22 minutes

~7 hours

>170000

amoA

~13 hours

4.Line 346

The discussion in this section introduced the selection of targeted databases. However, the reliability and advantages of the four databases in this paper have not described in discussion section. I suggest adding some description of the advantages for the database you choose.

Response: Thank you for your advice. We also believe that the more comprehensive the database is, the more sequences could designed primers covered. For napA gene database in our manuscript, we used the method of protein conserved domain alignment (HMMER) based on 472 napA sequences that were retrieved from GeoChip probe database. According to the E-value in HMMER (E < 10-5) and the annotation information of sequences, we selected 3397 highly credible sequences from the collected homologous sequences from NCBI. These sequences are with high reliability and complete description information. We added more description of the advantages of our database in the discussion section as your suggestion.

(Page 12, Line 346) “The design and assessment of primers are closely related to how reliable the sequence database is. Therefore, choosing a credible and complete database is very important for primer design. For napA gene database in our study, we used the method of protein conserved domain alignment to construct a database based on 472 napA sequences that were retrieved from GeoChip probe database. According to the E-value in HMMER and the annotation information of sequences, we selected 3397 highly credible sequences from the collected homologous sequences from NCBI. These sequences are with high reliability and complete description information.”

5.Line 335 Referring to the results of test data, we have created some rough statistics on the calculation time and data size (Table 6).

The description did not correspond with the table.

Response: Thanks for your correction, the table here has been deleted.

Reviewer 3 Report

In this paper the authors presented

a novel bioinformatic program able to design primers for microbial functional genes based on the k-mer algorithm, that bypassing the time-consuming step of sequence alignment, could greatly reduce run times while ensuring accuracy.”

The article is a short description of the program, the principles on which it is based, and describe the advantages and disadvantages of the new workflow described.

The article is well written, concise, and sufficiently clear, as far as this type of article may be.

I have tried to test all the workflow, and it seems to run correctly, at least with the dataset given as an example.

After solving some problems, related to the formatting of the file or the possible presence of hidden characters, the dataset has been recognized by the system and has been correctly analyzed.

The only problem seems to be the execution times, perhaps due to internet connection problems or data traffic and server commitment. The execution times are quite high and do not correspond with the times described in the article.

It is not possible to find a cause or a solution to this problem, but despite this, the approach used seems to be very interesting and noteworthy.

As also described in the article, practical amplification tests would be needed to verify the correctness of the results.

The final judgment turns out to be positive.

Author Response

Response: Thank you very much for your careful affirmation. As you say, due to the server connecting or hosting content is different, there may be some variation in the running time per test. Therefore, we try to test in the same environment as possible in our study. We will keep the website and program optimized and updated in future. And we are testing more experiments with ARDEP as well.